



**Surface deposition of marine fog and its treatment in the WRF model**
Peter A. Taylor[1], Zheqi Chen[1], Li Cheng[1], Soudeh Afsharian[1], Wensong Weng[1], George A.
Isaac[1,2], Terry W. Bullock[3], Yongsheng Chen[1]
[1] Centre for Research in Earth and Space Science, Lassonde School of Engineering, York University, Toronto,
Ontario, M3J 1P3, Canada
[2] Weather Impacts Consulting Incorporated, 20 Pine Ridge Trail, Barrie, Ontario, L4M 4Y8, Canada
[3] Met-Ocean & Digital Environment Solutions, 133 Crosbie Road, St. John's, NL, A1B 4A5, Canada
*Correspondence to:* Peter Taylor (pat@yorku.ca)
**Abstract** There have been many studies of marine fog, some using WRF and other models. Several model studies
report over-predictions of near surface liquid water content ($Qc$) leading to visibility estimates that are too low. This
study has found the same. One possible cause of this overestimation could be the treatment of a surface deposition
rate of fog droplets at the underlying water surface. Most models, including the Advanced Research Weather
Research and Forecasting (WRF-ARW) Model, available from the National Center for Atmospheric Research
(NCAR), take account of gravitational settling of cloud droplets throughout the domain and at the surface. However,
there should be an additional deposition as turbulence causes fog droplets to collide and coalesce with the water
surface. A water surface, or any wet surface, can then be an effective sink for fog water droplets. This process can be
parameterized as an additional deposition velocity with a model that could be based on a roughness length for water
droplets, $z_{0c}$, that may be significantly larger than the roughness length for water vapour, $z_{0q}$. This can be
implemented in WRF either as a variant of the Katata scheme for deposition to vegetation, or via direct
modifications in boundary-layer modules.

**1. Introduction**
This study was initiated when it was found that predicting fog in areas offshore from Atlantic Canada using the
NCAR/UCAR Weather Research and Forecasting model (WRF-ARW) was generally satisfactory in terms of fog
occurrence but gave high values of cloud water mixing ratio leading to visibilities that were too low compared to
observations. Other studies of marine fog had encountered similar problems (e.g. Chen et al 2020). Koračin et al
(2014) had noted "From the many modeling studies of sea fog, essentially numerical experiments/ simulations/
forecasting that started in the immediate post WWII period, it becomes clear that deterministic forecasting of sea fog
onset and its duration has generally been unsuccessful.". On land and over the sea the formation and decay of fog in
the atmospheric boundary layer is a complex issue involving many processes including cloud microphysics, long
wave and solar radiation, turbulent boundary layer mixing, advection and surface interactions. Modelling of fog, in
idealized one dimensional or single column models up to operational 3-D weather prediction and climate models is a
challenge which many have addressed over the years, as noted by Koračin (2017), Gultepe et al (2017) and many
others. Koračin et al (2014) review marine fog processes and studies up to 2014, noting the importance of air-sea
interactions. They discuss fog water deposition to vegetation extensively but not turbulent deposition to water





surfaces, and it is missing from their Fig 1 (and Fig 9.1 in Koračin 2017) showing " the main processes governing
the formation, evolution, and dissipation of marine fog". Although fog could be caused by mixing two slightly sub-
saturated air parcels and causing saturation due to curvature of the saturated mixing ratio versus temperature line,
most fog formation is initialized by cooling the lower parts of a column of moist, but unsaturated, air.  This can arise
because of long wave radiative heat loss from the underlying surface (radiation fog), vertical displacement of the air
column as it travels over sloping terrain or horizontal advection over a cooler surface. Our focus is on the advection
fog situation over ocean waters, a frequent occurrence over areas such as the Grand Banks and offshore areas of
Eastern Canada as the wind blows moist air from over the Gulf Stream towards the Labrador current (Taylor 1917;
Isaac et al 2020).

**1.1 Fog and the underlying surface**
The focus in this paper is on the interactions of fog water droplets with the underlying water surface, how this is
being modelled, how it could be improved in the widely used WRF model, and to briefly suggest some field
measurements to support this work. The basic hypothesis will be that, in addition to gravitational settling, turbulence
will induce collisions between fog droplets and the water surface and that most of these collisions will lead to
coalescence, so that the water surface is a sink for water droplets. This can be represented in terms of a deposition
velocity, over and above the settling or terminal velocity associated with small cloud droplets falling through air
under gravity and predictable assuming Stokes law (see, for example, Rogers and Yau 1989).  If there is an
enhanced turbulent deposition to the surface one would then expect the cloud droplet mixing ratio ($Qc$) to increase
with height ($z$) above the surface, at which $Qc$ would approach zero.  In a constant flux layer this would lead to a
logarithmic profile and allow the concept of a roughness length for cloud droplets, $z_{0c}$, although the profile can be
modified to incorporate gravitational settling. Not included is the possible creation of spray droplets by breaking
waves in high wind speeds, and this may need consideration in high seas with strong winds.

There have been many studies on the collision and coalescence of raindrops and cloud droplets, and of droplets
impacting hydrophobic surfaces but relatively few concerning interactions between cloud or fog droplets and ocean
surfaces. Over water the combination of wind and waves will lead to impacts occurring at a range of speeds and
incidence angles and relatively little is known about the details of this important interaction. The paper by Hallett
and Christensen (1984) and the reference to it by Isaac and Hallett (2005), although primarily on impacts at normal
incidence, do however support our expectation that fog droplets interacting with the ocean surface are likely to
coalesce eventually even if they may bounce on initial impact if that occurs at a shallow angle. If fog droplets do
collide with the underlying surface, whether it is the ocean, a lake, a water puddle on land or wet vegetation one
would expect coalescence and deposition of the fog droplets to the surface. Gravitational settling will play a role in
this but droplet impacts on the surface due to turbulence also need to be considered. As a result of deposition there
would be a reduction in the fog/cloud water mixing ratio ($Qc$), maybe to zero, at the lower boundary which would
lead to a positive value for $dQc/dz$ and a downward flux of $Qc$.





### 1.2 Aerosol and vegetation

If we broaden our view and consider aerosols in general, we find that significant work has been done in the same size range as fog droplets (1-50 μm). Recent reviews by Emerson et al (2020) and Farmer et al (2021) make it very clear that dry deposition (i.e. not rainfall related) of aerosol particles, solid or liquid, is a key process for their removal, that it is driven by turbulence and strongly dependent on particle size. For aerosol with diameters > 1 μm gravitational settling and turbulent diffusion both contribute to the overall deposition velocity. The aerosol studies include both water surfaces and vegetation. It is clear from Farmer et al (2021, Fig 3) that deposition velocity, $V_{dep}$, over water increases significantly with aerosol diameter between 1 and 50 μm, while this variation is somewhat less over other surfaces. Farmer et al's plots are not normalized by friction velocity or wind speed which probably accounts for some of the variability in $V_{dep}$ at fixed diameters.

There have been studies of fog deposition to vegetation and also to meshes designed to catch fog water (e.g. Section 3.4 of Gultepe et al 2017). However, as far as we are aware, the models of fog droplet deposition to water surfaces have either been via gravitational settling alone, ignored, or considered as a part of a turbulent, total water (vapour, $q$, plus liquid droplets) flux at the surface. Right at the surface the flux of water vapour will rely on molecular transfer alone while collision and coalescence of water droplets can be much more efficient and requires separate treatment.

### 2 Boundary-Layer modelling

For aerosols and sometimes other quantities, weather prediction, and other models tend to use deposition velocities ($V_{dep}$).to relate fluxes to an underlying surface to concentrations at some level above the surface. From a boundary-layer perspective, one often looks at the concentration profile and an eddy diffusivity. The simplest, and traditional, way to model flux-profile relationships of a quantity, s, in turbulent boundary-layer flow near rough walls is via an eddy viscosity/diffusivity, $K_s(z)=ku^*(z+z_{0s})$, where $k$ is the Karman constant (0.4) and $u^*$ is the friction velocity. The roughness length, $z_{0s}$, is specific to the property (horizontal velocity, temperature, mixing ratio ...) under consideration and will vary considerably depending on the physics of the final transfer process at the surface. The traditional way to determine $z_{0s}$ is to consider an approximately constant flux layer near the surface - leading to a logarithmic profile,

$$S - S_0 \approx (s^*/k) \log(z/z_{0s}), \tag{1}$$

where $S_0$ is the surface value. This will imply that $S = S_0$ at $z = z_{0s}$ and is the empirical way in which $z_{0s}$ can be determined. It is well known, see for example Garratt (1992, p 89) or Brutsaert (1982, p 121) that roughness lengths for momentum ($z_{0m}$) and heat or water vapour ($z_{0T}$, $z_{0q}$) transfers differ because form drag on roughness elements is the major cause of momentum transfer while molecular diffusivity at the surface is needed to effect heat transfer. As a result, $z_{0m} >> z_{0q}$, except maybe over very smooth aerodynamically smooth surfaces. We will propose the use of $z_{0c}$ for cloud droplet collision and coalescence with the water surface. We have no measurement data to determine a value, which might well vary with droplet size and sea state but can use reported aerosol studies to provide some guidance. We do however expect that $z_{0c} >> z_{0q}$.






If the fog has continued for some time one might expect that the relative humidity, $RH = 100\%$ in the fog layer, with
no significant condensation or evaporation. There will then be a near steady state in the lower fog layers with
constant downward $Qc$ flux ($F_{Qc}$). This flux will be a combination of turbulent diffusion and gravitational settling
($w_s Qc$) where $w_s$ is the gravitational settling velocity, based on Stokes law. If, as we will assume, Qc → 0 as z → 0
then turbulent transfer will dominate as the surface is approached and logarithmic $Qc$ profiles should result.
In our model calculations, with an eddy diffusivity, $K_c(z) = ku*(z+z_{0c})$, we find $RH \approx 100\%$ in the fog layers,
typically up to around 100m, and see constant flux layers with near-logarithmic $Qc$ profiles through most of this
height range, as in Fig 4. Departures from logarithmic are due in part to the effects of gravitational settling which
accounts for part of the downward flux.

Marine fog in the areas under consideration is often in moderate and high wind conditions (Isaac et al, 2020) and
relatively low heights ($<$ 10m) are used as the lowest model level. In that lowest, constant flux, "wall" layer with
neutral stratification, we can assume horizontal homogeneity, a constant downward flux of $Qc$ and a steady state.
We can then seek the solution to
$$w_s Qc + (ku*(z + z_{0c})\, dQc/dz = F_{Qc} = u*q_c*, \qquad (2)$$
where $F_{Qc}$ is a downward flux of cloud droplet liquid water mixing ratio. With $Qc = Qc_0$ at $z = 0$, the solution is,
$$Qc(z) - Qc_0 = (u*q_c*/w_s)\, [1\text{-}exp(-w_s\zeta/(ku*))], \text{ where } \zeta = \ln\left((z+z_{0c})/z_{0c}\right). \qquad (3)$$
If $w_s/u*$ is small, then to first order in $w_s\zeta/ku*$, (3) becomes simply
$$Qc(z)\text{-} Qc_0 = (q_c*/k) \ln\left((z+z_{0c})/z_{0c}\right), \text{ with } Qc = Qc_0 \text{ at } z = 0. \qquad (4)$$
If this is used to relate $z_{0c}$ to $V_{dep}$, and with $Qc_0 = 0$ we would have
$$V_d = u*k/(\ln((z_1+z_{0c})/z_{0c}), \qquad (5)$$
where $z_1$ is the height above the surface where $Qc$ is measured. This logarithmic profile approximation could be fit
to measured $Qc$ profiles to determine $z_{0c}$ from observations. As with $z_{0m}$ this is a somewhat empirical approach. In
the same way that the use of the $z_{0m}$ concept is widely accepted without precise calculation of the form drag on
roughness elements we would hope that future experimental determination of $z_{0c}$ would be a way to account for the
effects of turbulent collision and coalescence of fog droplets with a water surface. For radiation fog in low wind
speeds over land, stable air density stratification effects could be significant and can be accounted for with Monin-
Obukhov similarity modifications to $K_c(z,L)$ if the Obukhov length ($L$) can be determined.

The expected values of terminal velocity, $w_s$ for a droplet of diameter, $d$, and density $\rho$, falling under gravity ($g$)
through air of density $\rho_a$ and molecular viscosity, $\mu$, should be considered. In reality the fog droplet size distribution
will be broad and often bimodal (see Isaac et al 2020). The two peaks in some of Isaac et al's measured PDFs are at
diameters near 6 μm and 25 μm with Stokes law terminal velocities ($w = gd^2(\rho\text{-}\rho_a)/\mu$) of 0.001 ms$^{-1}$ and 0.019 ms$^{-1}$.
These are clearly small compared to wind speed but for the larger diameter, where the bulk of the liquid water
content (LWC) is often measured, the terminal velocity corresponds to 67 m per hour and will represent a
considerable removal rate in fog which may last several days. The key parameter in our constant flux with





gravitational settling model is $S = w_s/ku*$. In moderate winds over the ocean one might expect $u*$ values in the 0.1-
0.5 ms$^{-1}$ range, $k = 0.4$ and so the parameter, $S$ will generally be in the range 0.006 to 0.46 while $\zeta$ may be 5-10 at the
lowest grid point, implying that gravitational settling can play a significant role and that Eq. (3) may provide a more
appropriate profile for the larger droplets. In principle Eq. (3) should be used to refine any $z_{0c}$ estimates from
measurements. For typical friction velocities (0.1 - 0.5 ms$^{-1}$) and with the lowest model level at $z_1 = 1.7$ m with $z_{0c} =$
0.01 or 0.001 m, $V_d$ values would be in the range 0.005 to 0.04 ms$^{-1}$, quite comparable with the gravitational settling
velocities so both will play a role in the modelling of deposition to the surface.

Ideally values for $z_{0c}$ would be established from field measurements BUT we are not aware of any height profiles of
$Qc$ in fog over water and for now will treat $z_{0c}$ as a tuning parameter in our models. Over most land surfaces, $z_{0m}$ is
considered independent of Reynolds number and we might hope that the same would apply for $z_{0c}$. Over water
surfaces with ripples and waves as the roughness elements life gets more complicated and the roughness length for
momentum, $z_{0m}$, can be wind speed dependent, governed by the Charnock-Ellison relationship[1] (Charnock 1955),
$z_{0m}=au*^2/g$, where $a$ is referred to as Charnock's constant, with typical values in the 0.01 - 0.03 range. Establishing
precise over water values for $z_{0c}$ will prove at least as difficult as for $z_{0m}$, noting that it may also vary with droplet
size, but it does provide a framework for representing this potentially important fog deposition process.

**3. Past Field and Laboratory Measurements**
There have been many field measurements in marine fog, including, notably, G.I. Taylor's (1917) work over the
Grand Banks, and more recently the C-Fog study reported by Fernando et al (2021). As far as we are aware none
have provided the $Qc(z)$ profile data from which we could make $z_{0c}$ determinations.

Over land there are some multi-level $Qc$ measurements indicating lower values near ground than above. Also lower
droplet numbers. Kunkel (1984) reports measurements of advection fog in July 1980 and July 1981, at 2 levels (5m
and 30m) on a tower "in the middle of a large, flat, open area" about 12 km inland from the Atlantic on Cape Cod.
There is some variability but his liquid water content values ($W$, gm$^{-3}$) are always higher at 30m than at 5m and the
ratios are generally between 2 and 3. There are some differences in droplet size between the levels but they are
relatively modest and less consistent. Ignoring stratification effects, assuming that a logarithmic profile is
appropriate and that $Qc_0 = 0$ then the ratios of 2 and 3 in $Qc$ correspond to $z_{0c}$ values of 0.833 m and 2.04 m. If $Qc_0$
were > 0, say some fraction of $Qc(5m)$, then the $z_{0c}$ values would be higher. Pinnick et al (1978) report $Qc$
measurements, from February 1976 above an inland site in Germany, at multiple heights up to 180 m with light
scattering instruments carried aloft by a tethered balloon. Water content was calculated from particle size
distributions and, from their photographs, the local land surface appears open and flat. Their sample profiles, in fog
and haze, generally show $Qc$ increasing with height and 3 of 4 cases shown are consistent with increases by factors
of 2-3 between 5 - 30 m. Most of their results appear to be in radiation fog with light wind conditions. Klemm et al

---

[1] Henry Charnock always told me that Tom Ellison had suggested the dimensional analysis behind what is generally referred to as the Charnock relationship, so I refer to it in this way. - Peter Taylor



(2005) report eddy covariance measurements of fog water fluxes to a spruce forest at Waldstein, in a mountainous
area of Bavaria Germany, and compare results with related model studies. They report that "turbulent exchange
...dominates over sedimentation at that site" and investigate relationships between liquid water content ($LWC$, gm$^{-3}$)
and visibility. Their flux model is based on a deposition velocity, $V_{dep}$, with deposition to the canopy,
$F_{tot} = V_{dep} * LWC$ , including both turbulent flux and gravitational settling. They note that some studies at the same
location (Burkhard et al, 2002) report significant differences in downward flux at different levels (flux at 22m can be
45% less than at 35m), perhaps illustrating the difficulty of making representative measurements close to the canopy
top. Evaporation of fog droplets is also cited as a possible cause of these differences. It is perhaps also worth adding
that fog water collectors (e.g. Schemenauer and Cereceda, 1991) can enhance the amount of fog water that is
removed at ground level and provide an important source of clean water for some isolated communities. a removal
efficiency of 20% is estimated for a 2-layer, 12m x 4m polypropylene mesh.

Turning to aerosol studies, Farmer et al (2021) provide an extensive list of laboratory and field studies of aerosol
deposition to both land (grassland, forest, snow and ice) and water surfaces. Many provide $V_{dep}$ values for aerosols in
our size range. Deposition velocity measurements in wind tunnel studies in a short report by Schmel and Sutter
(1974) are interesting, but lack details of how the aerosol flux to the surface was determined. From their Fig 3 we
can estimate average deposition velocities for selected particle sixes and wind speeds. Unfortunately, it is not clear
at what heights their wind speeds were measured and their $z_{0m}$ and $u*$ values are somewhat suspect. If we assume
that $z_{0m} = 0.0002$ m and that wind speeds in their tunnel were measured at a height of 0.1 m then their average U (7.2
ms$^{-1}$) and $u*$ (0.44 ms$^{-1}$) values are reasonably consistent and their $V_{dep}$ value of 0.04 ms$^{-1}$ for 6 μm diameter aerosol
would lead to $z_{0c} \sim 10^{-4}$ m. For larger diameter aerosol (28 μm) $V_{dep} = 0.37$ ms$^{-1}$ and $z_{0c} \sim 0.062$ m with the same
wind assumptions, suggesting strong size effects, but we are wary of suggesting precise values.

Field data studies in the Farmer et al (2021) list include studies on Lake Michigan by Caffrey et al (1998) and Zufall
et al (1998) with deposition to surrogate surfaces, and a recent report by Qi et al (2020) from the NW Pacific Ocean.
These and other papers confirm the strong size dependence of deposition velocity and acknowledge wind speed
dependence but are often concerned with long term estimates of the deposition of chemical species to the ocean or
lake rather than short term events. One way in which wind speed plays a role is via wave breaking and "broken"
water surfaces, a concept used in a model proposed by Williams (1982). This proposes that dry deposition of aerosol
particles is considerable different between smooth and broken patches of the water surface with a much higher
resistance over the smooth areas.

To briefly summarize we believe that there are observations to support the idea that the underlying land or water
surface can be an effective sink for fog droplets, and other, similar sized, aerosol. The deposition velocity will have
a dependence on droplet size, especially over water, but there is a lack of reliable data, even over land, to calibrate
our simple, roughness length based approach to modelling the turbulent deposition of fog droplets. Our roughness





length, $z_{0c}$, will have to remain as a tuning parameter until more extensive fog droplet profile and flux measurements
can be made.

**4 Model Studies**
As reported by Koračin (2017), there have been many studies aimed at understanding and/or predicting the
occurrence of fog, and Kim and Yum (2012) also provide a review focused on marine fog. For our purposes it is
relevant to see how different model papers discuss deposition of fog water to the surface and their surface boundary
conditions on $Qc$. The model of Brown and Roach (1976) focusses on radiation fog, in relatively low wind speeds
and provides an excellent summary of the key components needed to model fog formation and its life cycle,
including radiation, turbulent diffusion and gravitational settling. They note that " liquid water (as well as water
vapour) is also lost to the ground by turbulent diffusion and gravitational settling of droplets." and their lower
boundary conditions include $w = 0$ for $z = 0$ and $t > 0$, where $w$ is their liquid water mixing ratio. Brown and Roach
assert that "$K_h$ , $K_q$ , $K_w$, exchange coefficients for heat, water vapour and liquid water respectively" are assumed
equal in their model. In adiabatic conditions they state $K = kzu*$ but avoid discussion of roughness length.
Extrapolating their $w$ vs log $z$ profiles to $w = 0$ would indicate a $z_{0c}$ value, for liquid water, of slightly less than $10^{-2}$
m. This is consistent with their use of the $K$ model of Zdunkowski and Barr (1972) who set $z_0 = 1$ cm. Zdunkowski
and Barr's treatment of the conservation equation and lower boundary condition for $M$, the total moisture content
(vapor plus droplets), plus zero flux of $M$ to the surface, generally leads, inappropriately, to liquid water profiles
with maxima at the surface. Barker (1977) developed a similar model for maritime boundary-layer fog and also uses
the same eddy diffusivity and roughness length for heat, water vapour and liquid water. He assumes that cloud liquid
water concentration (his $l_0$) is zero at the water surface.

The COBEL and COBEL–ISBA 1-D models developed in France (Bergot 1993; Bergot and Guedalia

1994; Bergot et al 2005), have been used successfully at Paris's Charles de Gaulle International Airport. Bergot and
Guedalia (1994, hereafter referred to as BG) provide details of dew and frost deposition to the underlying surface
and note its importance. However their dew flux is based on direct condensation of water vapour to the surface (BG
Eq 22) as the inverse situation of evaporation. Their liquid water ($q_t$) diffuses and has a gravitational settling
velocity (BG Eq 17, 18) but no surface condition is specified and one assumes that the only flux to the surface is
through gravitational settling. Few details are given on the surface boundary conditions in the latest journal
publications but contour plots, e.g. Fig 13c from Bergot et al (2005) generally show $Qc$ maxima at the surface.
COBEL has also been coupled with WRF (Stolaki et al 2012) and used to simulate advection-radiation fog
conditions at Thessaloniki's airport.

Bott and Trautmann (2002) proposed PAFOG as "a new efficient model of radiation fog" and it has been

used by others, including, recently, and coupled to WRF, in a study by Kim et al (2020). PAFOG is a 1- dimensional
($z,t$) model developed as a more practical version of the more complete MIFOG model (Bott et al 1990) which
carries multiple aerosol and size bins for fog droplets. The MIFOG model includes dynamics and thermodynamics
but focusses on interactions of radiation (solar and long wave) with fog droplets of varying size. The cloud droplets
that evolve in the model have a bimodal size distribution which varies with time with large droplets descending





under gravity, and being removed at the surface, at a faster rate than the small ones. The dynamics include turbulent
mixing via eddy diffusivities for momentum and heat. Water droplet number concentrations in each size bin are also
subject to diffusion with the same diffusivity as heat. The diffusivities are given by Forkel et al (1987). It appears
that a common roughness length, $z_0 = 0.05$m, is used for momentum, heat and water droplets. No boundary
conditions are given in Bott et al (1990) but from the results presented it would appear that there is no turbulent flux
to the surface, only deposition via gravitational settling in MIFOG. The same appears to be true with PAFOG apart
from possible removal of cloud water by vegetation as described by Siebert at al (1992a,b). PAFOG appears to give
good results for 2-m visibility (Bott and Trautmann 2002, Fig. 1). Their Fig. 2 generally shows high $Qc$ values (0.2,
0.3 g kg$^{-1}$) extending almost down to the surface but with a sudden drop near $z = 0$ in 3 of the 4 contour figures
shown. There is similar near-surface behavior of $Qc$ in Siebert's results but it is not clear why. All of the above
papers have a lack of detail on surface boundary conditions.

Shuttleworth (1977) and later Lovett (1984) were early modelers of fog deposition to vegetation, using

resistance concepts ($1/V_d$). Katata et al (2008) later developed a land surface model (mod-SOLVEG) including fog
and cloud water deposition on vegetation and on forests. The downward flux of cloud water is due to both turbulent
mixing and gravitational settling (Katata 2014) and Katata et al (2008) successfully compare their model predictions
with field measurements from a forest site near Waldstein in Germany. The turbulent fluxes use a vertical eddy
diffusivity, $K_z$, and multiple vegetation levels are involved. They claim that their model results compare well in
comparison with Klemm et al.'s (2005) application of the Lovett (1984) model. Lovett points out that there can be
"turbulent transfer of cloud droplets to the canopy" and that, in windy conditions "inertial impaction is the dominant
mechanism". These model papers all deal with forests and Katata et al (2011) describe the implementation of the
ideas within WRF using the MYNN 2.5 Planetary Boundary Layer scheme and WSM6 cloud microphysics. The
central assumption is that, within, what Katata et al (2011) call org-WRF, fog water deposition to the surface can be
represented as,

$$F_{Qc} = C_h |\underline{U}| \rho Q c = V_d \, \rho Q c \qquad (6)$$

where $\underline{U}$ is the wind vector at the lowest model level and $\rho$ is air density. $C_h$ is a bulk transfer coefficient for height $h$
above the surface (specifically the lowest model level, although $h$ was later defined as the canopy height), $V_d$ is a
deposition velocity, associated with turbulent diffusion but including gravitational settling.  In what Katata et al
(2011) call fog-WRF the deposition velocity is set to

$$V_d = A/U, \text{ where } A = 0.0164(LAI/h)^{-0.5}, \qquad (7)$$

Here $LAI$ is leaf area index (m$^2$ per m$^2$) and here $h$ is canopy height (in m). so that the coefficient 0.0164 has units of
m$^{0.5}$. Values given for $A$ in Katata et al (2008) for both needle leaf and broad leaf trees are mostly in the range 0.02 -
0.04. with $U$ measured "over the canopy". If the $U$ and $Qc$ measurement height was at 10 m, $Q_C(z_{0c}) = 0$  and $z_0 = z_{0c}$
$= 0.1$m then, from Eq (2) and the log wind profile, $A = 0.0075$, but with $z_0 = z_{0c} = 1$ m the result is $A = 0.03$, in the
middle of Katata's range.

Recent papers by Wainwright and Richter (2021) and Richter et al (2021) focus on marine fog using a large

eddy simulation model, following on from the work of Maronga and Bosveld (2017) and Schwenkel and Maronga
(2019, 2020) on LES studies of radiation fog. The marine fog models use Morrison et al (2005) microphysics. The



cloud water ($Qc$) and cloud droplet number ($Nc$) equations include turbulent diffusion and sedimentation but there
seems to be no enhanced deposition to the surface. Most results (e.g. Figs 3a, 6, 10, and most of Fig. 11 from
Wainwright and Richter 2021) appear to show $Qc$ maxima at the surface although Fig.7 in Schwenkel and Maronga
(2019) suggests a rapid drop in $Qc$ near the surface. There seems to be little discussion of deposition of fog droplets
to the surface in most of these papers although, for their Lagrangian simulations, Richter et al (2021) note " At the
bottom of the domain, droplets that hit the water surface are removed from the simulation, and a new super-droplet
is immediately introduced randomly in the domain according to the same procedure for initialization." It is not clear
what this does in terms of a flux to the surface but their results (Fig 3 of their paper) in a simulation of advection fog
show number densities that are maximum at the fog top, around 30 m after 10 h, while $Qc$ and mean droplet radius
are maximum near the ground.

None of the papers that we have found use the $z_{0c}$ approach that we have adopted, although the resistance

and deposition velocity ideas of Lovett (1984) and Katata et al (2008) are closely related. When roughness lengths
are used, the values for $Qc$ always appear to be the same as for water vapour.

### 312    4. Operational NWP models

Fog forecasts have been a challenge for operational NWP models as indicated by many authors including Wilkinson
et al (2013) who note the Gultepe et al (2006) opinion that " most NWP models were unable to provide accurate
visibility forecasts, unless they accounted for both liquid water content and droplet number." We also note the
comment of Bergot et al (2007), "Current NWP models poorly forecast the life cycle of fog, and improved NWP
models are needed before improving the prediction of fog".

Wilkinson et al (2013) focus on the droplet number issue and, in a somewhat "ad hoc" fashion, the UK Met

Office Unified Model at that time applied "a taper curve for cloud droplets near the surface." This reduces droplet
numbers between the surface and 150m without changing liquid water concentration. Droplets are then larger, have
higher settling velocities and so " the impact ... is greatest closest to the surface, where they increase the amount of
($Qc$) removed from the lowest model levels." It seemed to work but their "taper curve" approach could certainly be
considered somewhat "unphysical".

Yang et al (2010) made an evaluation on the Canadian GEM-LAM model for marine fog off the east coast

of Canada with nesting down to 2.5 km, using both visibility reports and Qc comparisons with observed
measurements from the FRAM project (Gultepe et al 2009). Three case studies are presented with the overall
conclusion that GEM-LAM forecasts at 2.5 km resolution underestimate $Qc$ and had a warm and dry mean bias at
the lowest model level. This is opposite to our WRF studies which predict high $Qc$ values at low levels. An earlier
evaluation by de la Fuente et al (2007) had reported that, "... It has been shown that the current operational 15 km
regional GEM forecast is insufficient for forecasting (sea) fog." The GEM-HRDPS (Milbrandt et al 2016) uses a
MoisTKE treatment of the boundary layer which is described in Belair et al (2005). It works with the variable $q_w =$
$q_v + q_c$ , where $q_c$ is the total cloud water content (droplets + ice fragments) which is mixed vertically using an eddy
diffusivity $K_H$, as for heat. Assuming that surface transfers are of $q_w$ this suggests no special treatment of cloud
droplets over water surfaces. Milbrandt et al (2016) indicate that the cloud microphysics then used in GEM-HRDPS



were based on MY2, the two-moment bulk microphysics scheme described in Milbrandt and Yau (2005). That paper
includes the statement "... because cloud droplets are assumed to have negligible terminal fall velocity." Fall speeds
were given for different hydrometeor categories but not for fog droplets. As discussed above, terminal velocities
under gravitational settling are small (mm s$^{-1}$), and can probably be considered negligible in a convective cloud but
for long lasting marine fog they can play an important role. Currently GEM-HRDPS uses P3 microphysics
(Morrison and Milbrandt, 2015). This includes gravitational settling of cloud droplets but there are subtle
distinctions between explicit and implicit $q_c$ from the microphysics and the boundary-layer treatments and there
appears to be no surface flux of $q_c$, just a flux of $q_v$.
Teixeira (1999) reported on ECMWF successes in fog forecasting at that time with the Tiedtke (1993)
cloud scheme forecasting liquid water content. The Musson-Genon (1987) surface boundary-layer treatment treats
diffusion of total water with a low surface roughness length, but includes gravitational settling of liquid water.
Teixeira's conclusions include the statement " The comparison between the simulated and the observed visibility
shows that the onset of fog, the lowest values of visibility and the dissipation stage are properly simulated." In terms
of marine fog in the Grand Banks area the reanalysis data showed that "The comparison between the model's fog
climatology and the climatological data shows that the model is able to reproduce most of the major fog areas,
particularly over the ocean." The ECMWF (2020) model physics are documented at https://www.ecmwf.int/en/
elibrary/19748-part-iv-physical-processes, with Chapter 3 giving information on interactions with the surface. As in
our approach their transfer coefficients involve roughness lengths. Over water they specify $z_{0m}$, based on the
Charnock-Ellison relationship plus a laminar flow value based on molecular viscosity ($v$), while for moisture they
specify $z_{0q} = \alpha_q v/u*$, with $\alpha_q = 0.62$ (from Brutsaert, 1982), assuming simply molecular diffusion in a viscous
sublayer. It is important to note that the ECMWF model deals with total water as a conservative variable, $q_t = q + q_c$
$+ q_i$, and that $z_{0q}$ thus applies to water vapour, water droplets and ice fragments. The subscript "$t$" seems to be lost
after Eq 3.3 in the ECMWF document but we assume that in what follows from that point, e.g. in their Eq. 3.6, $q =$
$q_t$. Over land there are some adjustments but over water fluxes are proportional to ($q_n-q_{surf}$) where $q_n$ is at the lowest
model level and $q_{surf}$ is the surface value. The values of $q_{surf}$ is set to 0.98 $q_{sat}(T_{sk})$, where $T_{sk}$ is the water surface
"skin" temperature, implying that surface relative humidity is close to 100% AND that $q_c \approx q_i \approx 0$. This
approximately agrees with our conjecture BUT the ECMWF model assumes the same $z_0$ for water vapour and cloud
droplets while our conjecture is that $z_{0c} >> z_{0q}$. There is gravitational settling, with terminal velocities, $v_x(D)$, for rain
and snow (their Eq 7.20, 7.21) but not for cloud droplets.
In the USA there are many different forecast models but we will just consider the Rapid Refresh (RAP) and
High Resolution Rapid Refresh (HRRR) Models, based on WRF-ARW, (Skamarock et al 2019). These are run
operationally, with 13 km and 3km resolution meshes by NCEP and NOAA/ESRL Global Systems Laboratory.
They use the same MYNN boundary-layer and Thompson microphysics modules as in our coastal fog simulations
and thus may have similar limitations in depositing fog droplets over water. Going back to a statement in Zhou and
Du (2010), "Although one hopes that the liquid water content (LWC) at the lowest model level can be explicitly
used as fog, experience indicates that an LWC-only approach does not work well with the current NWP models due
mainly to two reasons: one is the too coarse model spatial resolution and the other is a lack of sophisticated fog





physics." Things have changed since then but the recent "somewhat improved" statement (including the qualifier,
somewhat) on visibility performance by Alexander et al (2020) can be noted.

**5. Fog deposition treatment in the WRF model with module_bl_mynn and module_sf_fogdes**
WRF versions 4.1.2 and 4.2.1 (https://www2.mmm.ucar.edu/wrf/users/downloads.html), and possibly earlier
versions, march forward in time with separate modules for dynamical and multiple physical processes (see
Skamarock et al 2019; Olson et al 2019). For the benefit of readers familiar with, or interested in, the WRF model
we provide some details, here, in Section 6 and in Cheng et al (2021). The WRF modules used here treat
gravitational settling and turbulent diffusion as separate processes and compute separate tendencies, including
deposition rates. Gravitational settling is included within the Thompson microphysics module and Eq. (4) is used to
compute deposition velocities associated with turbulent diffusion with $V_d = u^*k/(\ln((z_1+z_{0c})/z_{0c})$, where $z_1$ is the first
Qc model level above the surface. The surface boundary layer is treated in a 1-D implicit finite difference mode with
tridiagonal matrices set up for turbulent kinetic energy, velocity components, potential temperature, humidity and
cloud liquid water $Qc$. Variables are defined at the centres of grid cells with fluxes at the upper and lower
boundaries. For the cells adjacent to the ground the fluxes at the upper cell surface use an eddy diffusivity ($K$)
approach, which for a downward flux of cloud water is of the form $K(Qc(2)-Qc(1))/dz$ where $Qc(1)$ is the value in
the centre of the lowest level grid cell and $dz$ is the vertical separation. The turbulent flux to the lower boundary, in
this case the water surface, is computed with a deposition velocity. For cloud water the (negative) upward flux is
*flqc* and is computed in module_bl_mynn as *-vdfg (Qc(1)-sqcg)* with the deposition velocity $V_d = vdfg$ provided by
module_sf_fogdes and with $Qc$ on the surface, $sqcg = 0$. In the unmodified module_sf_fogdes, water surfaces are
classified as "other" and the deposition velocity assumed is just the settling velocity of the cloud droplet falling
through air under gravity. In a turbulent flow over a wavy water surface the deposition velocity should also include
the effects of turbulence bringing droplets to impact the water surface and coalesce, and *vdfg* should be higher.
There are different ways in which this can be implemented in WRF module_bl_mynn (see Cheng et al, 2021).

**6. WRF SCM set-up and tests**
As a basic test of our treatment of deposition of fog droplets to a water surface and for comparisons against the
regular WRF schemes we use the single column version of WRF (em scm xy), one of the ideal test cases described
by Skamarock et al (2019). In our applications of the SCM we used several boundary layer and microphysics
schemes, set up various vertical grids with up to 201 levels, and different lowest and upper levels. Initial soundings
have close to 100% relative humidity in the lowest few hundred meters, moderate wind speeds typical of the NW
Atlantic and WRF-SCM was typically run for 36 - 84 h. To simplify interpretation of the results, our SCM runs are
without any solar or long wave radiation. Surface temperatures were cooled for several hours and then held steady.
The main interest is to see the impact of fog droplet deposition to the underlying water surface. Physics and
Dynamics components of the WRF namelist input are listed in Cheng et al (2021). Turbulent deposition to the
surface is represented via a deposition velocity, $V_d$, multiplying the lowest level $Qc$ value at $z = z_1$. This is set as

$$V_d = ku^* / \ln (1+z_1/z_{0c}), \tag{7}$$



where $u*$ is the friction velocity, $k$ (= 0.4) is the Karman constant and $z_{0c}$ is a roughness length specific to water
droplets diffusing to a water surface and coalescing. In principle it could be dependent on sea state and droplet size.
As noted above, roughness lengths can represent different processes for turbulent transfers of heat, water vapour,
momentum, and fog droplets of liquid water to the surface, and should not all be the same. Our assumption is that $z_{0c}$
(for fog/cloud droplets) should be significantly larger than $z_{0q}$ for water vapour.

WRF-SCM was run using modules bl_mynn, for boundary-layer turbulent transfers, and mp_thompson, for cloud
microphysics, to generate the results shown in Figs 1-3. Since gravitational settling is represented within
mp_thompson the parameter grav_settling was set to 0 in bl_mynn (see Olson et al, 2019, section 6.4). No radiation
effects are included. Lack of long wave radiation will affect mixing at the top of the fog layer but we will focus on
lower boundary issues. In the results below the initial sounding has potential temperature of 300 K at the surface
increasing with height at a rate of 4 K km$^{-1}$. The initial relative humidity was 100 % at the surface dropping to 0 at 6
km.  The wind profile was established with a long, no cooling run and has a geostrophic wind of (20,0) ms$^{-1}$. Sea
surface temperature was cooled at a rate of 3 K h$^{-1}$ for 6 h and then held fixed. The lower boundary condition
included a flux of water droplets to the surface, computed with a deposition velocity determined by Equation (8)
above and using a range of $z_{0c}$ values.

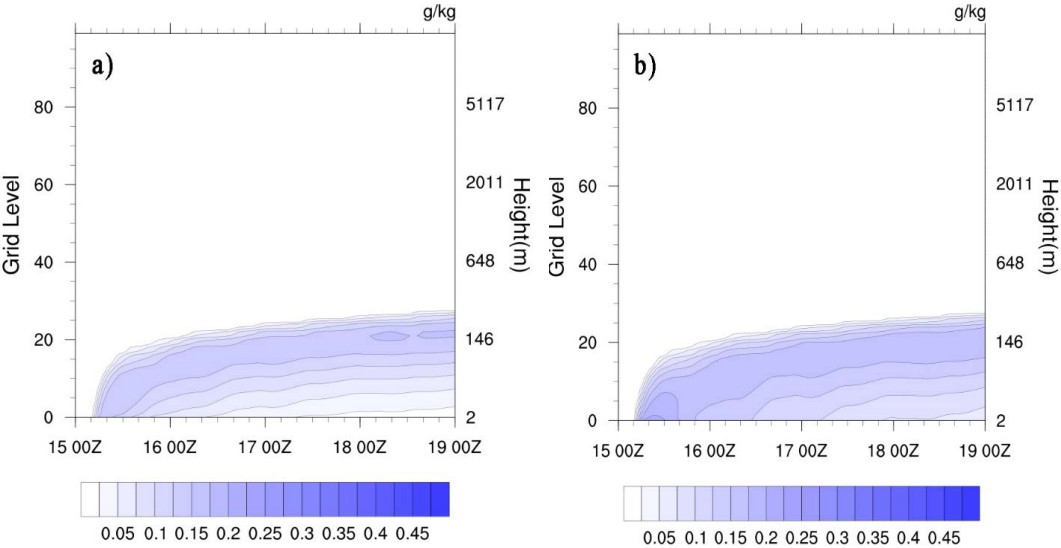



**Figure 1: Contours of $Qc$ (g kg$^{-1}$) generated by WRF SCM with 6h of surface cooling at 3 K h$^{-1}$ a) MYNN boundary layer**
**using the turbulence deposition scheme described with $z_{0c}$ = 0.01 m plus Thompson microphysics with gravitational**
**settling, b) Original MYNN module with gravitational settling only in Thompson microphysics. The full vertical domain is**
**shown to indicate that no upper level cloud formed in these cases - it did with other input. Times on the x axis are in the**
**format DD HHZ, with small tic marks 4 hours apart. Run start time was 15 00Z.**






Fig. 1 shows contours of $Qc$ (g kg$^{-1}$) as it varies with ($t$, eta grid level) from the model calculations over 4 days
starting, somewhat arbitrarily, at 00Z on day 15 of a month (15 00Z) so that cooling runs to 15 06Z. Some height
levels are marked to indicate the grid stretching in $z$. These runs are for latitude 44° N (Sable Island) with 101 eta
grid levels. The WRF model operates with a sigma type vertical coordinate ($\eta$), decreasing from 1 at the lower
boundary to 0 at the upper boundary, where $p = p_t$. It has a simple form over a flat surface. Details are in Skamarock
et al (2019). Our model grid points are not uniformly spaced in $\eta$ and the spacing increases smoothly with increasing
height (decreasing $\eta$). We set $p_t \approx 22000$ Pa to give a top boundary at about 12 km. The Eta levels start at $\eta =1$ (the
surface) decreasing to $\eta = 0$ and $p = p_t$ at Eta level 101 (our SCM model top). In full 3D runs we take $p_t = 5000$ Pa.
The grid is staggered so that variables like $T, Qv, Qc, U, V$ are at mid-levels while the lower boundary ($z =0$) is at the
base of the lowest grid cell. Our 'grid levels" start with the centre of the lowest cell (0) and increase upwards. In Fig.
1a, $z_{0c} = 0.01$ m while Fig. 1b is for results with the original MYNN scheme with no surface deposition except for
gravitational settling in the Thompson microphysics. Fog forms as a result of the surface cooling and extends from
the surface to around eta level 20, which corresponds to $z \approx 150$ m. We were initially concerned by the wave-like
features in the contour lines. These have a period of around 17 h and arise because of inertial oscillations (of period
$2\pi/f$) in the wind field as it adjusts to the cooling of the surface and changing turbulent momentum transfers. They
decay slowly as the wind profile adjusts to the cooler surface. Values of $Qc$ are lower in Fig. 1a because of turbulent
deposition to the surface. Fig. 2 shows Qc profiles with the MYNN boundary layer, 24 h after the start of the model
calculations and 18 h after the end of surface cooling. The additional turbulent deposition can play an important role
in lowering $Qc$ levels in the boundary layer while, in this case, not having a significant impact above 100m. The
amount of the reduction depends on the value chosen for $z_{0c}$.

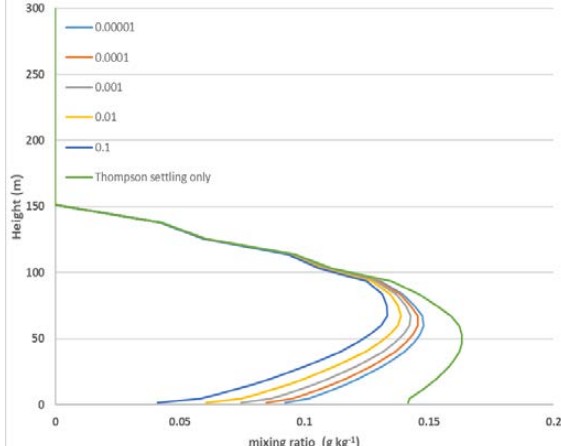



**Figure 2: Qc profiles 24 h after the start of the integration and 18 h after the end of the surface cooling, by 18 K. Results**
**with the original MYNN (gravitational settling in Thompson microphysics only) and with a range of $z_{0c}$ values (in m).**
**Time step, $dt = 60$ s, 101 levels.**






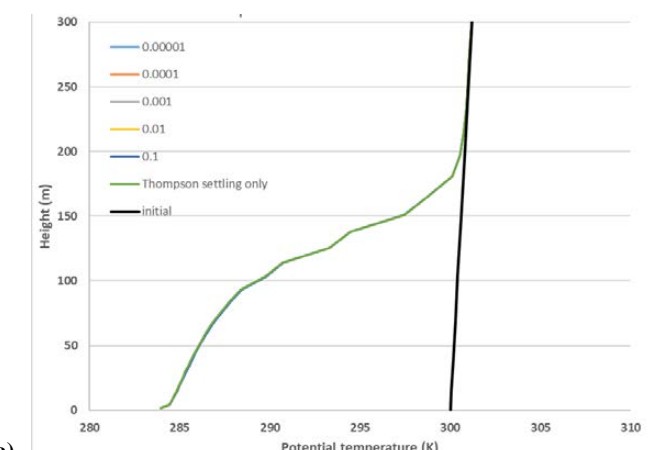

**a)**

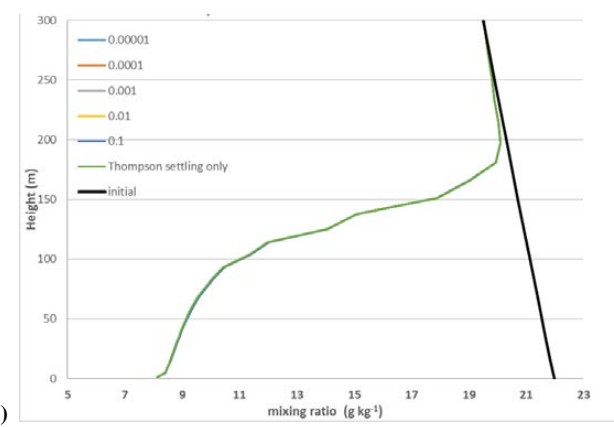

**b)**

**Figure 3: a) Potential temperature ($\theta$) and b) $Q_v$ profiles corresponding to Fig. 2, including the initial profiles. Note $z_{0c}$**
**deposition of cloud droplets has minimal impact, all curves overlay.**

It is interesting to note that the removal of $Q_c$ at the lower boundary has minimal impact on the predicted
temperature and water vapour, $Q_v$ profiles (Fig. 3). It could however be important when fog starts to evaporate if the
air temperature rises. Note that in generating these results we have not included radiation (short wave or long wave)
effects in order to focus on the impacts of turbulent deposition at the water surface. Radiation can play a significant
role once fog has formed, and in particular long wave radiational cooling at the fog top (Yang and Gao, 2020) can
add to the cooling rate and can enhance turbulent mixing in the upper part of the fog layer. The center of the lowest
grid layer is at 1.7 m. Noting the "kinks" in the profiles at the lowest level in profiles of $Q_c$, $Q_v$ and $\theta$, we
investigated possible causes and plotted them on an expanded height scale (not shown). They arise because in WRF
modules sf_mynn and sf_fogdes the fluxes to the surface are computed with deposition velocities involving


$\ln((z+z_0)/z_0)$ while the eddy diffusivities used to compute fluxes at the top of the first level and levels above are
based on length scales proportional to $kz$ without the $z_0$ addition. This will not be significant for $z \gg z_0$ but with
the lowest computational levels close to the surface this could be modified.  This is an internal WRF issue, noted in
comments in the bl_mynn module code.

A further point from Fig 3b is that with our near saturated initial profile and strong cooling there is a significant
reduction in $Qv$, of order 10 g kg$^{-1}$ throughout the lowest 100 m. This will be converted to $Qc$ but after 24 h most
will have been deposited to surface, through both gravitational settling, as in the "original" curves in Fig. 2, or by a
combination of gravitational settling and turbulent deposition to the water surface as in the other cases shown in Fig.
2. In runs with gravitational settling turned off (not shown for this case but see Fig 4b) and no turbulent deposition
the $Qc$ values increase significantly, to around 6 g kg$^{-1}$ near the surface after 12 h. Gravitation settling prevents very
high values from occurring but additional turbulence induced deposition further limits them.

**7. 3D test cases**
Turning to the 3D WRF model we have been running the model for North Atlantic simulations for summer 2018 on
a domain extending from eastern Canada out beyond the Grand Banks and including Sable Island. A separate paper
on comparisons with visibility measurements on Sable Island is in preparation. These 3D runs have no additional
surface cooling and are simply run as hindcasts of the actual situation with initial and boundary conditions taken
from NCEP analyses. The sea surface temperatures are held fixed for daily 36 h runs, generally with a 12 h spin up.
Note that the input initial and boundary fields had zero $Qc$. They are run with hybrid_opt = 0, and in the vertical
direction we have a straight sigma coordinate, $\eta = (p_d - p_s)/(p_t - p_s)$ with $p_t$ = 5000 Pa. Runs were also made with
hybrid opt = 2 and Qc results were almost identical. Solar and long wave radiation can use either Goddard or
RRTMG scheme and we used the MYNN PBL scheme with both Thompson and the WSM6 microphysics options.
For details of these options see Skamarock et al (2019). Fig. 4 shows sample results from 6 h runs with the full 3D
model using Thompson microphysics and Goddard radiation, long and short wave. They show a similar response to
the SCM (Fig. 2) when turbulent deposition of cloud water to the surface is introduced. The top figure (4a) shows a
normal run with the Thompson microphysics module accounting for gravitational settling effects. MYNN has
turbulent deposition to the surface but no gravitational settling (grav_settling = 0).  In the lower figure (4b) we
removed gravitational settling from the Thompson microphysics scheme (av_c = 0) as well as from MYNN. With no
turbulent deposition to the surface, and, in one special case with no gravitational settling either, there are higher $Qc$
values as expected. These 3-D runs used NCEP analyses as initial conditions but the initial $Qc$ was set to zero
everywhere. In fog the analysis would give 100 % RH and the model then generated $Qc$ within a few hours but
without the strong temperature and $Qv$ drops that were simulated in our SCM tests. Gravitational settling (Fig. 4a)
has reduced the peak $Qc$ values at around 100 and 900 m from the case with no settling and the $Qc$ removed from
those levels has settled and mixed downwards to increase the $Qc$ values near the ground. Additional 3D runs were
made with the standard MYNN codes and the Katata scheme using modified deposition velocities in the "other"
case. These matched our results obtained with a modified MYNN code. Also, in place of the Thompson






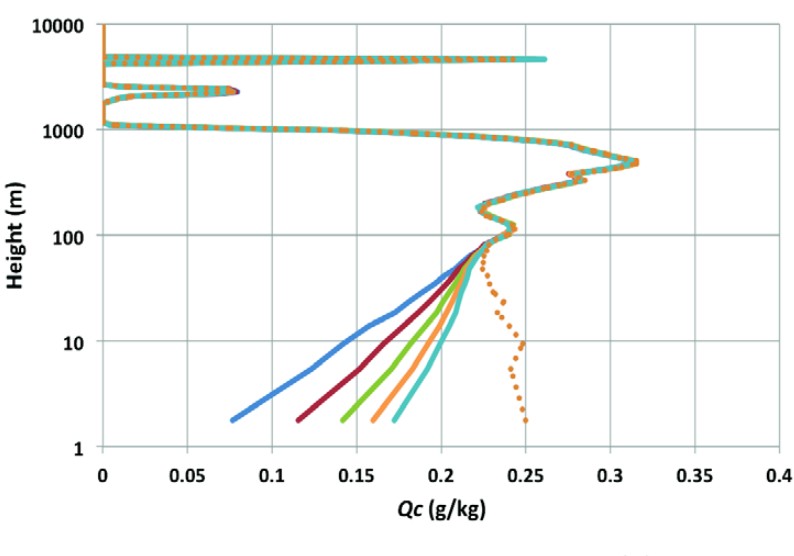

a)

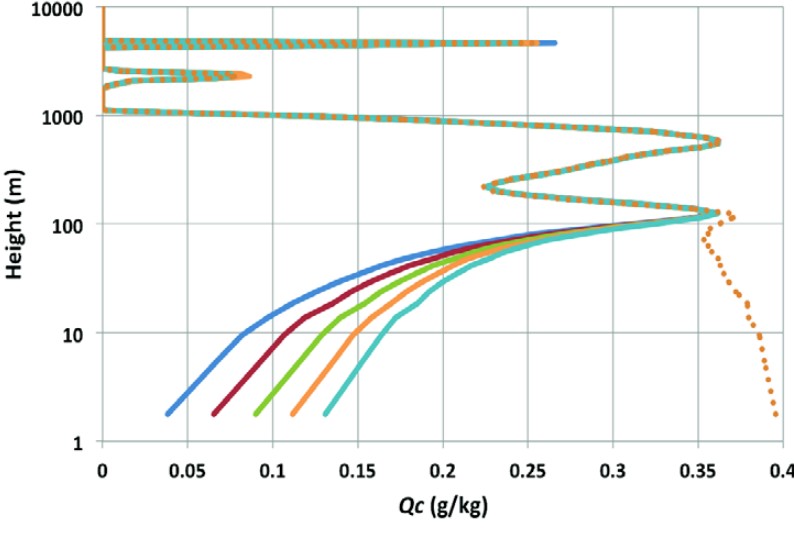

b)

**Figure 4: Sample 3-D WRF output at a fixed location over the Grand Banks, with different $z_{0c}$ values (given in m) in $Qc$**
**turbulent deposition, a) with and b) without gravitational settling. Start time was 7/1 12Z, 2018 and results are for 7/1**
**18Z. Results are with MYNN boundary layer and Thompson microphysics.**


microphysics scheme we ran tests with WSM6 microphysics. In all cases there was a large impact of turbulent
surface deposition of $Qc$ in the lowest 100 m, even with very low values for $z_{0c}$. As an initial guide we suggest using
$z_{0c} = 0.01$m or 0.001m as a modest value which has a solid impact. We should also emphasize that gravitational
settling also has an impact on $Qc$ values near the surface and both processes need to be included in models.

**8 Visibility considerations**

Models can predict liquid water mixing ratios but the critical forecast issue is visibility which will depend on the
number and size distribution of the fog droplets. In dense marine fog ($LWC > 0.05$ g m$^{-3}$), Isaac et al (2020, Fig. 12)
show that the size distribution of marine fog droplets is generally broad and frequently bimodal, raising concerns
about all simple diagnostic schemes. Despite such concerns, models such as the one proposed by Isaac et al (2020)
assume that visibility, or Meteorological Optical Range, $MOR$ is proportional to liquid water density, $LWC$ (g m$^{-3}$ or
kg m$^{-3}$) or mixing ratio (g kg$^{-1}$ or kg kg$^{-1}$), $LWC^{-2/3}$ times $N^{-1/3}$ where $N$ is the droplet number density (m$^{-3}$). Some
models include dynamic equations for $N$ while others assume prescribed values, typically $N = 10^8$ m$^{-3}$. If the size
distribution were well known and universal this could work but as Isaac et al (2020) note the size distribution in fog
over the ocean can be bimodal and the number density can vary widely. In conditions with air density x $Qc > 0.005$
g m$^{-3}$ the number density reported by Isaac et al over a site in the Grand Banks area varies between $10^7$ and $3\times10^8$
m$^{-3}$. Medians were close to $N = 0.8\times10^8$ m$^{-3}$. Note however that these measurements were at a height of 69 m above
the ocean surface and if the water surface is a sink for cloud droplets one would expect lower values, and maybe a
different size distribution, at the WMO standard visibility measurement height of 2.5 m (WMO, 2020). Chen et al
(2020) note problems with too low visibility from their WRF calculations coupled to the Kunkel (1984) visibility
equation ($vis = - ln(\varepsilon)/\beta$ with the extinction coefficient (km$^{-1}$), $\beta = 144.7$ $W^{0.88}$ where $W$ is in g m$^{-3}$). The contrast
threshold, $\varepsilon$ was given as 0.02 by Kunkel but is set to 0.05, as recommended by the WMO (Boudala et al 2012; Chen
et al 2020). In the GSD algorithm used in NCEP's Unified Post Processor version 2.2, the Kunkel result is used with
$\varepsilon = 0.02$ for visibility reductions in clouds, plus additional effects of aerosol, rainfall and humidity. The relationship
between visibility or $MOR$ and $Qc$ or $W$ can vary in these models between a power of -2/3, through -0.88 to -1 if $N$
were proportional to $Qc$, but all show that too high a value of $Qc$ will lead to too much reduction in visibility.
Running standard versions of WRF one can compute visibilities with either the Isaac et al (2020) equations or the
GSD algorithm used in NCEP's Unified Post Processor version 2.2 (for details, see Lin et al 2017). Both led to
significantly lower values of $MOR$ than were reported on Sable Island. Typical WRF values being of order 1/10 -
1/5 of the reported visibility, suggesting $Qc$ values that may be high by a factor between 5 and 30. Visibility - cloud
water relationships are open to revision, with different values of ε and noting the scatter in Isaac et al's (2020) data,
but there is a strong suggestion that WRF values of $Qc$ are too high without adding additional $Qc$ deposition. Fig. 5
shows sample visibility time series computed from 3D WRF $Qc$ output for Sable Island, interpolated to $z = 2$m, for
two 36 h periods in 2018 when fog was reported at that location. Original WRF runs with just gravitational settling
show seriously limited visibility ( < 100m) on some occasions when METAR visibility was closer to 1 km while
with added turbulent $Qc$ deposition and a range of $z_{0c}$ values, the optical range was a better match to the
observations. These are sample cases and a more extensive comparison is planned.

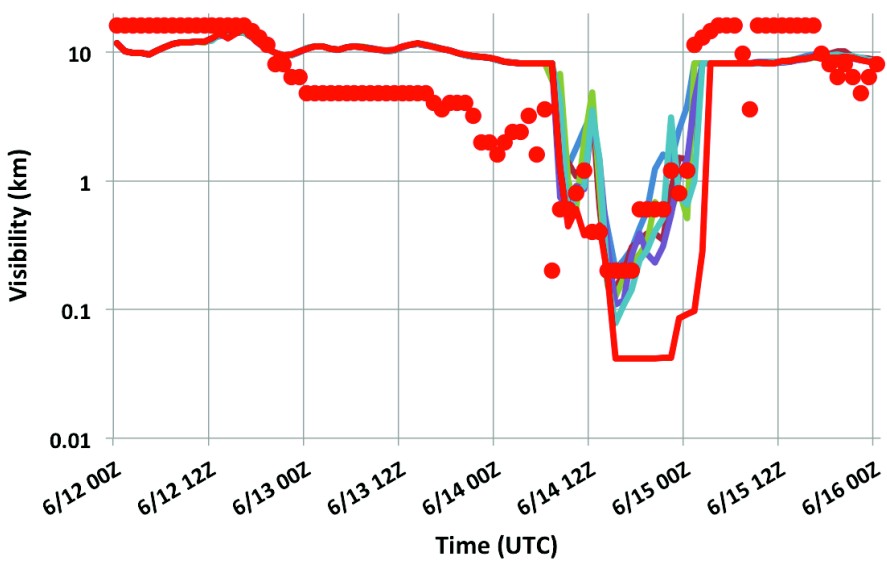

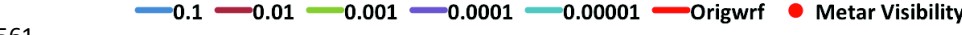


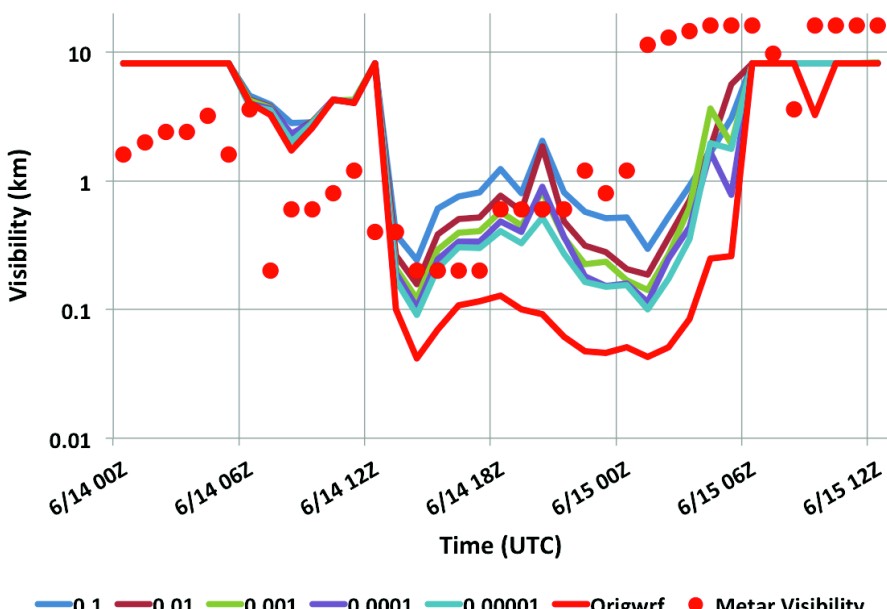


**Figure 5: Sample June 2018 GSD visibility hindcasts for Sable Island at 2m, using MYNN boundary layer and WSM6**
**microphysics. with different $z_{0c}$ values, given in m.**



**9. Conclusions**
It has been known for many years that fog water can be deposited on vegetation and this has been incorporated into
some boundary-layer fog models. It is also known that μm size aerosols can be removed from the atmosphere by
turbulence at water, and other, surfaces (Farmer et al, 2021). It then seems surprising that, for marine fog, turbulence
induced cloud/fog droplet deposition to water surfaces has not been recognised by most modellers as a significant
potential addition to the deposition associated with gravitational settling. Neglecting this can then lead to fog liquid
water mixing ratios being too high and visibility forecasts being too low. This applies to specialised boundary layer
models and to numerical weather prediction models. Many authors have noted the difficulties and complexity of
modelling fog and accurately forecasting visibility. Getting everything right will be extremely challenging but, for
marine fog, recognising that a significant process is missing from many models could be a step in the right direction.
WRF-ARW is a major contribution to the atmospheric research endeavour and the developers and
maintainers of this huge, multi-faceted, publicly available model deserve huge credit. As with anything of this size
and complexity, developed and modified over many years by many individuals, it can be very hard for new users to
trace through the source codes and understand just how they work. Some module codes are well documented and
commented, others less so. Running the model is made relatively easy, and it is designed to be robust. We have done
our best to understand some details and ensure that our modifications, briefly explained in Cheng et al (2021), do
what we expect but we make no guarantees!
Based on our modelling of marine fog with WRF, and reviews of the treatment of boundary layer fog in
WRF and other models, it seems that a much better understanding of fog droplet interaction with the ocean surface,
and other surfaces, is needed. Laboratory studies might be possible, and numerical simulations, but with some good
in situ profile measurements through fog layers over land and water one could start to better understand and
parameterize this process. Any foggy location on land could work but Sable Island would offer an ideal location for
such a study in marine fog. It is a 43 km long, narrow (mostly < 2 km wide) sand bar in the Atlantic Ocean about
175 km offshore from Nova Scotia, Canada. It has some vegetation, cranberry bushes and grass, wild horses and
many seals and is now a National Park. An upper air station (CWSA, 71600) was operated there by Environment
Canada until August 2019.  The western tip of the island could be an ideal location for a tall mast with a variety of
fog related and standard meteorological research instrumentation at multiple levels. Observations
(https://climate.weather.gc.ca/climate_normals/index_e.html) show more than 200 (out of 720) hours of fog
(visibility < 1 km) on Sable Island in the months of June and July.  Taylor et al (1993) made use of Sable Island as
an accessible offshore platform to study frontal passages over the sea in winter during the Canadian Atlantic Storms
Program (CASP 86). Summer 2022 could be a good time to return.

**Code availability**
WRF codes used are readily available from  https://github.com/wrf-model/WRF/releases/tag/v4.2.2 . Modifications
and additional details are in Cheng et al (2021).

**Author contributions**





ZC ,LC, SA, WW and YC were primarily involved in aspects of the WRF code adaptation and model runs. PAT,
GAI and TWB were primarily involved in reviewing background information and interpretation of results. PAT
prepared the manuscript with contributions from all co-authors.

**Competing interests**
The authors declare that they have no conflict of interest.

**Acknowledgements**
Financial support for this research, for which we are very grateful, has come primarily through a Canadian NSERC
Collaborative Research and Development grant program (High Resolution Modelling of Weather over the Grand
Banks) with Wood Environmental and Infrastructure Solutions as the industrial partner. Initial support was also
through Peter Taylor's NSERC Discovery grant. We would like to thank Anton Beljaars for providing guidance and
many valuable comments as well as Ayrton Zadra and Jason Milbrandt for their help in tracking down details of
Environment Canada's GEM model. Trevor VandenBoer pointed us to the aerosol work and Joe Fernando allowed
two of us to attend a C-Fog meeting in 2019 where we also had useful discussions with Will Perrie and Rachel
Chang.

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

particulate species to surrogate surfaces deployed in southern Lake Michigan. Environ. Sci. Technol. 32:1623–
28

---------------------------------------