# Peer review of "Cheng, L., Chen, Z. and Chen, Y.,"

_Atmospheric Chemistry and Physics, 2021_

## Referee Comment (RC2)

**Review of the paper entitled**
**Surface deposition of marine fog and its treatment in WRF model**

by Peter Allan Taylor et al.

**manuscript number ACP-2021-344**

This paper addresses the difficult topic to investigate the effect of turbulent transfer of fog droplets to the surface. Current NWP models do not take into account this process leading to an overestimation of fog liquid water content near the ground. This model bias is well known but has not been studied in detail with a scientific approach like in this article. This work studied the fog deposition through the Monin-Obukhov framework and a roughness length for liquid water. To my knowledge, this is the first time that this kind of approach is used to parametrized fog deposition. The proposed parameterization is tested in 1D and 3D WRF simulations, illustrating the impact of fog deposition on vertical profiles of liquid water content. This work is innovative and very interesting.

However, I think that some shortcomings should be addressed before the manuscript would be acceptable for publication. Therefore, I can not recommend publication of this paper without major revisions. I recommend that the paper be accepted conditionally with modifications detailed below.

1. **effect of fog deposition on fog life cycle** :
   This article focusses only on vertical profiles of liquid water content (eg fig 1, 2, 4). In my opinion, this approach is necessary but is too restrictive. The impact of fog deposition on fog life cycle and on fog properties should also be studied in detail.
   -What is the impact of deposition on LWP? And consequently what is the impact on radiative processes?
   -What is the impact of deposition on fog life cycle? At which stage (formation, mature, dissipation) is the fog most sensitive to deposition? A time series comparing the amount of water collected at ground by droplet settling and/or by deposition would be helpful to better understand the influence of both processes.
   -What is the influence of wind on fog deposition? The geostrophic wind used in this study is relatively strong, about 20 $m/s$.
   Please elaborate.

2. **effect of fog deposition on fog spatial extension** :
   In 3D simulations (section 4), I would have liked to see horizontal extension of fog with and without deposition on surface. Does the deposition have an influence on the horizontal extension of fog? on advective processes? What is the impact of deposition on horizontal heterogeneities of the fog layer? Please elaborate.

3. **visibility considerations** :
   In my opinion, the section 8 "Visibility considerations" is out of the goal of this article and does not improve the scientific merit of this article. Numerous visibility parameterizations have been done (eg Kunkel, Gultepe) showing a big dispersion of results. I agree that the visibility parameterizations are very sensitive to the liquid

water content, but it is a well-known results and your study does not have added values in this domain. I would have preferred a discussion on the effect of deposition on fog properties (see previous remarks) and/or a discussion section illustrating the perspective of your work.

4. **notations** :
   -section numbering : 4 model studies (l227) and 4 Operational NWP models (l312)
   -numerous notations for liquid water content : $Q_c$ (eg l61) , $W$ (eg l178) , $LWC$ (eg l192) , $q_c$ (eg l332)
   -reference to Zhang's work : Boundary-Layer Meteorol (2014) 151:293315 and http://cerea.enpc.fr/fich/these_zhang.pdf

---

## Author Comment (AC1)

**Re: manuscript number ACP-2021-344 -RC2 - Response**

We appreciate Dr Bergot's comments and suggestions, noting his extensive experience with fog, covering the Paris-Fog field program and many modelling studies. The comments on life cycle, spatial extension and visibility are addressed below.

The life cycle of fog is of special concern in situations on land where one is generally concerned with radiation fog with a strong diurnal cycle. Most of our attention in this paper is on marine fog where, as Isaac et al (2020) have shown, there is essentially no diurnal cycle. We do try to use Sable Island visibility data for comparison with our model predictions (Fig 5) on the basis that this is essentially a marine situation. Our subsequent work, using WRF with higher spatial resolution around Sable Island (Cheng et al, 2021) has revealed a diurnal cycle and differences with the surrounding ocean.

A column of marine advection fog can probably last for several days as it travels over cooling water and there will be some evolution. With our 1-D tests in Fig. 1 we do show some 4-day results as our column evolves in a case with initial (6 h) cooling followed by fixed surface water temperature. Our main goal in the paper has been to raise the issue of enhanced surface deposition of fog droplets to a water surface, over and above that caused by gravitational settling. This is in part to promote the measurement of vertical profiles of fog so that we, and others can better understand and represent the process.

In a separate research note Peter Taylor (2021) is looking into deposition velocity issues, of fog and other aerosol through models of "Constant Flux Layers with Gravitational Settling". This will address issues of which process, gravitational settling ($w_s$ or $V_g$ are used as symbols) or turbulent flux and deposition, carries the downward flux. Typically, both processes are important with different ratios at different levels. A critical parameter in this division is $S = w_s/ku_*$ , as in Eq (3). In strong wind speeds S will be small and Eq (4) suggests that gravitational settling is less important, at low levels. In low wind speeds, typical of radiation fog on land, and if S is relatively large, say O(1) then gravitational settling becomes more important.

In our 1-D simulations we have used relatively high geostrophic winds, but high winds are not unusual in marine fog over the Grand Banks (Isaac et al, 2020). In our 3-D simulations a range of wind speeds occur. We have run WRF for 3 months (June, July and August) in 2018, 36 h runs with 12 h spin-up each day. Comparisons are being made with data from Sable Island (Chen et al, 2021). Wind speeds at 10 m are mostly < 15 ms$^{-1}$ but geostrophic winds of 20 ms$^{-1}$ are quite common.

With regard to spatial extension, with our 3-D WRF simulations we always look at plots and animations of Qc over our d02 domain (see Cheng et al, 2021) at the lowest model level. Fig R1 is an example of a 2D plot of Qc at the lowest model level at the same time as Fig 4a in the paper. The black dot identifies the Grand Banks location used in Fig 4. The value of $z_{0c}$ was 0.01 m in Fig R1a while in R1b there is no turbulent deposition, just gravitational settling. The importance of gravitational settling, as in Fig 4b, can be seen from Fig R2. In this case gravitation settling has been turned off in the Thompson microphysics module and leads to some relatively high Qc values in some areas when there is also no turbulent deposition.

We could have included additional figures like those below, but we were trying to keep the paper at a reasonable length.

[Figure]

Figure R1  2D fog plots at 7/1, 18Z, 2018. Thompson microphysics with gravitational deposition, a) z0c = 0.01 m, b) no turbulent deposition, related to Fig. 4, a);

[Figure]

Figure R2.  2D fog plots at 7/1, 18Z, 2018.  Thompson microphysics without gravitational deposition, a) z0c = 0.01 m, b) no turbulent deposition, related to Fig. 4 b).

We included Section 8 on visibility since the forecasting of visibility is the main practical application of fog forecasting and we wish to demonstrate the potential utility of more accurate forecasts of Qc at low levels. We certainly agree that we need to model more than just the liquid water content, needing to know fog droplet numbers and, ideally, size distributions, in order to better estimate optical range. For now, however the Isaac et al (2020) parameterization that we used was based on theory, and we used a constant droplet number concentration ($10^8 m^{-3}$), compatible with measurements from the Grand Banks area where our model improvements could be applied.   We chose to test its performance, in a very preliminary way, and Chen et al (2021) will expand on these comparisons using Sable Island visibility data.

We can try and expand a little on our conclusions but the main message we want to get across is that, apart from Katata et al's (2011) work over forests, most fog models have under-estimated fog water removal via turbulent deposition to the underlying surface. We have suggested and tested a logical, boundary-layer, approach to enhance surface deposition over water, and possibly other surfaces, but it does involve an unknown parameter, $z_{0c}$. This needs determination from measured fog profiles. We hope that publication of this paper will encourage others, and ourselves, to get into some fog and measure many quantities in as much detail as possible. For marine fog, the Fatima project will be an important start (https://efmlab.nd.edu/research/Fatima/).

We appreciate the suggestion to study the paper by Zhang et al, 2014, but could not access the thesis. We are looking into the 2014 paper, especially for follow on work with radiation fog on land. Different authors use different notations and we will try to avoid use of too many variants, and will also renumber the sections.

Again, we appreciate the careful review and useful suggestions.

References

Zheqi Chen, Li Cheng, Peter Taylor and Yongsheng Chen, 2021, Simulating fog over Sable Island using the Weather Research and Forecast (WRF) model, in preparation

Li Cheng, Zheqi Chen, Peter Taylor, Yongsheng Chen and George Isaac, 2021, Fog over Sable Island, CMOS Bulletin, July 2021,  https://bulletin.cmos.ca/fog-over-sable-island/

Taylor, Peter A., 2021, Constant Flux Layers with Gravitational Settling: with links to aerosols, fog and deposition velocity. Submitted to ACP. (ACP-2021-594)

---

## Author Response (AR1)

Title: Surface deposition of marine fog and its treatment in the WRF model
Author(s): Peter Allan Taylor et al.
MS No.: acp-2021-344

Responses to referee comments RC1, RC2, were posted (AC1, AC2) on the interactive discussion web site, https://acp.copernicus.org/preprints/acp-2021-344/#discussion. We have no further responses.

RC3:  No response seemed needed. We focussed on citing Zhang et al (2014) rather than Zhang's 2010 thesis, although we have looked at it.

Revisions made in response to Referees.

RC1:  This is a very positive review and we have not made any significant changes based on it, except to reference the companion article (acp-2021-594) currently posted as preprint in ACPD. Specifically in lines 164, 165, we say " A more detailed analysis is presented in a companion ACP discussion paper, Taylor (2021).". This is in relation to the combined effects of gravitational settling and turbulent diffusion in the downward flux of fog water to the underlying surface.

RC2:  As noted in our detailed response, AC1: 'Reply on RC2', Peter A. Taylor, 14 Jul 2021, we have avoided too much discussion of "life cycle" but have looked at the spatial extent of the modelled fog, and have now added Fig 4 to illustrate "spatial extension" from our 3D WRF runs. We have also tried to avoid different notations for the same quantities. "We will use Qc for mixing ratio (g kg$^{-1}$ or kg kg$^{-1}$) and $LWC = \rho_a Qc$, where $\rho_a$ is air density, as liquid water content (kg m$^{-3}$ or g m$^{-3}$) unless discussing results from specific papers where, for clarity, it is sometimes useful to use their symbols."

We have followed up on the work undertaken by Meteo France and included reference to important work by Mazoyer et al (2017) and Zhang et al (2014) that we had overlooked, perhaps because our focus is on marine fog.

Other Revisions
Dr Isaac and I were participants in the recent ICCP on-line conference (https://iccp2020.tropmet.res.in/agenda). Based on talks and posters presented there, we have added material related to recent field programs, including LANFEX, SoFog on land and C-Fog (coastal and marine locations), and to the upcoming Fatima program on marine fog (https://efmlab.nd.edu/research/Fatima/).

We have done our best to match the word template.  I have not tracked all the format changes, including reference formats, or slight modifications to figures to avoid excessive tracking. I have however tracked text changes. There are a number of editorial revisions but no significant content changes apart from those noted above.